# Regional Adolescent Obesity and Overweight Status in Korea from 2018–2019: Comparison between Two Data Sources

**DOI:** 10.3390/healthcare9121751

**Published:** 2021-12-17

**Authors:** Dong-Hee Ryu, Su-Jin Lee

**Affiliations:** 1Department of Preventive Medicine, Daegu Catholic University School of Medicine, Daegu 42472, Korea; 2Cooperation Support Team, Daegu Public Health Policy Institute, Daegu 41940, Korea; iamssu1122@naver.com

**Keywords:** adolescent, geographical location, Korea, obesity, overweight, regional health planning, schoolchildren

## Abstract

Difference in the regional adolescent obesity level may cause a notable health inequality between regions since it significantly affects adulthood health status. This study examined 2018 and 2019 regional obesity and overweight status of adolescents aged 12 to 18 by comparing two cross sectional population-based data sources, the Korea Youth Risk Behavior Web-based Survey (KYRBS) and the National Student Health Examination (NSHE). Prevalence was estimated by calculating weighted percentages and 95% confidence intervals. Correlations in the relative rankings of each municipality were determined by computing Spearman correlation coefficients (*r_s_*), and prevalence discrepancies between the data sources were visualized by simple correlation graphs. The geographical distributions of adolescent obesity and overweight status showed no perfect concordances between the data sources regardless of sexes and survey years. For adolescent obesity status, there were significant difference between the least and the most obese regions and *r_s_* levels were fair to good with *p*-values less than 0.05, but the correlation graphs indicated body mass index (BMI) underreporting in the KYRBS. For adolescent overweight status, no significant similarities were defined between the data. These results can be used as a basis for the establishment of related policies.

## 1. Introduction

According to the World Health Organization (WHO), global pediatric obesity or overweight prevalence increased from 4% in 1975 to 18% in 2016 [1]. Similarly, recent obesity and overweight trends in Korea are also increasing from 15.3% in 2007 to 23.7% in 2017 [2]. Adolescent obesity in Korea is known to be associated with many risk factors including low level of fruit and vegetable intake, consumption of ramen, poor sleep hygiene, physical inactivity, and depression [3].

Obesity in adolescence is a complex matter that is affected by both genetic and non-genetic factors [4]. While most earlier studies focus on genetic factors, researchers in recent years have begun to shift their attention toward examining obesity’s relationship with socio-environmental factors. Several review articles have reported that the identification and modification of socio-environmental influences that promote less daily physical activity and the excessive consumption of low nutrition foods among adolescents is necessary [5,6,7]. One of the most common ways studies have considered socio-environmental factors regarding this issue in Korea is by examining the association through categorizing regions into urban (large cities and medium- or small-sized cities) and rural areas [8,9,10]. A major problem with these studies is that the characteristics of local environments classified based on city size or urbanity are treated the same within the group, meaning that considering the distinctive characteristics of each region is impossible. Other research interpreted the magnitude and trends of the adolescent obesity burden at regional levels by using a single data source collected at the national level [11,12,13]. The practical outcomes of policies proposed and implemented based on the conclusions of such studies are most likely not effective due to their simple and superficial data interpretations. Even though a few studies based on the determinant levels of the Socio-Ecological Model [14] have also been conducted in Korea by using a single cross sectional data source collected at the national level [15,16] or by using a longitudinal data composed of a small-sized cohort [17], studies examining the issue at the regional level are still scarce. Meanwhile, news articles usually provide the national statistics [18] or report relative rankings of regions based on overweight or obesity percentages [19]. The advantage of this reporting tactic is that readers can easily grasp the contexts, but there is still a validity issue.

The most primitive and simple unit that divides a country into regions according to their distinctive characteristics is a municipality. Not only geographic but also economic, fiscal, and social characteristics vary according to administrative districts, which eventually leads to differences in the degrees of developments or advancements in the regions. Such variabilities between regions are easily observed in South Korea since almost half of its population resides in a few areas: specialized area called the Seoul Capital Area [20] and, areas in and around the capital city (Seoul, SU), including one of the metropolitan cities (Incheon, IC) and a provincial area (Gyeonggi, GG) (Figure 1). The Seoul Capital Area is Korea’s financial, political, and cultural center as well as its most modernized region. The metropolitan cities (Busan, BS; Daegu, DG; Daejeon, DJ; Gwangju, GJ; and Ulsan, US) are large cities that are less advanced than the Seoul Capital Area but more modernized than provincial areas. Sejong (SJ), the only Special Self-Governing City, is not a metropolitan city but has unique characteristics because it was newly launched in 2012. The provinces (Gangwon, GW; Chungbuk, CB; Chungnam, CN; Gyeongbuk, GB; Gyeongnam, GN; Jeonbuk, JB; and Jeonnam, JN) mostly include rural areas, and their population is lower than metropolitan cities’. The only Special Self-Governing Province, Jeju (JJ), is the one island area among the seventeen municipalities and its regional characteristics differ from other areas’ since its main industry is tourism. Each municipality has its own political, cultural/sub-cultural, and even linguistic characteristics, and more importantly, the levels of fiscal availabilities are also different due to municipalities’ varying budget management and execution scales. Hence, it is necessary to consider administrative districts and their characteristics rather than using a regional factor that is classified by its size or urbanity when approaching the adolescent obesity issue. 

Adolescent obesity is significantly associated with an increased risk of severe obesity in adulthood [21], where a person would likely develop multiple acute or chronic diseases requiring greater healthcare needs later in life [21] and a reduction in life expectancy [22]. A systematic review pointed out that obesity during adolescence is the best predictor of obesity in adulthood [23]. Therefore, the current level of adolescent obesity in each region should be considered in the aspect of health inequality between regions in the future. Since strategies for the issue require a prevention approach [5], the establishment of relevant related policies is necessary to help produce positive results, including the enhancement of health-related behaviors and the reduction in diseases, disabilities, or unexpected deaths in the adulthood. 

In this study, two sets of well-known, cross sectional, and large-scale nationally representative data collected from students within educational infrastructure, the Korea Youth Risk Behavior Web-based Survey (KYRBS) and the National Student Health Examination (NSHE), were assessed at the regional levels. In the KYRBS, students were asked to self-report their latest height and weight while, in the NSHE, they were directly measured. The purpose of this study was to examine regional obesity and overweight status with the two data sources and to evaluate whether applying the subsequent results in the policy-making processes associated with adolescent obesity and overweight status at regional levels is reasonable. 

## 2. Materials and Methods

### 2.1. Materials

In this study, the 2018 and 2019 KYRBS and NSHE were evaluated due to the following reasons. 

The growth chart was newly revised in 2017 and became publicly available in 2018. In the revised version, the 95th percentile limit for boys’ body mass index (BMI) was adjusted since it was found that it was previously set too high [2].An analysis of single-year data may result in coincidental conclusions.

As mentioned above, there is a difference in the methods used for measuring height and weight between the KRYBS (self-reported) and the NSHE (directly measured). Instructions regarding height and weight measures are noted in Article 4 of the School Health Examination Regulations. There were no changes in reporting or measuring methods for both survey years.

#### 2.1.1. KYRBS

The KYRBS is a nationally representative study annually conducted by the Korea Disease Control and Prevention Agency (KDCA, formerly known as the Korea Center for Disease Control and Prevention). Participants aged 12–18 years are selected using a stratified multistage probability sampling method. Students from 800 schools (400 junior high and 400 high schools) anonymously complete self-administered questionnaires using computers at schools. All participants provided written informed consent and detailed information regarding the survey is available elsewhere [24]. The KYRBS focuses on examining health-risk behaviors of adolescents including the following: tobacco, alcohol, and substance use, dietary behaviors, weight control efforts, physical activity, mental health, sexual behaviors, and socioeconomic status. The number of students who participated in the 2018 wave was 60,040 (response rate: 95.6%), and 57,303 students (response rate: 95.3%) participated in the 2019 wave. 

#### 2.1.2. NSHE

According to Article 7 of the School Health Act, the head of a school must arrange health examinations to protect and promote students’ health [2]. Heads of schools select medical institutions for student health check-ups, and the services provided include the following: checking of physical developmental status and the presence/absence of acute or chronic diseases, measurement of blood pressure, urine tests, oral tests, X-ray, and laboratory tests related to obesity (blood sugar, total cholesterol, and liver enzyme tests for obesity high-risk individuals). The clinical tests, including the measurement of blood pressure, urine test, oral test, X-ray, and laboratory tests, are only provided to 1st, 4th, 7th, and 10th graders. Participants are also asked to fill out survey questionnaire forms. The results of the health examinations of selected subjects are collected using a digital educational information system called the national education information system (NEIS). The NSHE selects sample schools using stratified multistage probability sampling. Detailed information about the NSHE is available elsewhere [2]. The number of students from the sampled schools who checked physical developmental status in the 2018 wave was 107,954 and 104,380 students in the 2019 wave. To compare obesity and the overweight status data of adolescents with that of the KYRBS data, data from elementary school students (1st–6th grades aged from 6 to 11) were not analyzed in this study. 

### 2.2. Statistical Analysis

BMI was calculated as weight (kg) divided by the squared height (m^2^). According to the 2017 revised Growth Chart, adolescents with BMI ≥ 95th percentile were defined as obese and those with BMI ≥ 85th percentile and <95th percentile were defined as overweight. The mean and standard deviation (SD) of weight were calculated for each 2 cm height interval, and subjects who did not fall within the mean ± 5SD weight ranges were excluded from the analysis to minimize misclassification due to input errors. Students whose age were inaccurate or missing were also excluded.

Obesity and overweight prevalence were estimated by calculating weighted percentages and 95% confidence intervals (CI). To examine concordance in the geographic pattern of the prevalence between the data sources, Spearman correlation coefficients for relative rankings were computed. Simple correlation graphs were used to find the degree of numerical agreement between the data sources. To make judgments about relative rankings, up to three decimal points were considered, although this study’s results only provided up to one decimal point. All statistical analyses were performed using SPSS version 19.0 (BMI, Armonk, NY, USA) and *p*-values of <0.05 were considered to indicate statistical significance. Graphics were produced using Microsoft Excel (Microsoft Corporation, Redmond, WA, USA).

### 2.3. Ethical Statement

The ethical evaluation was exempted since publicly available data were used. 

## 3. Results

The total number of the KYRBS (self-reported) and NSHE (measured) participants included in the analysis were 114,083 (58,336 for the 2018 wave; 55,747 for the 2019 wave) and 134,079 (68,861 for the 2018 wave; 65,218 for the 2019 wave), respectively (Figure 2). For both data sources, the study participants’ mean age was about 15 (Table 1), ranging from 12–18.

### 3.1. Adolescent Obesity Prevalence

Not surprisingly, the overall obesity prevalence derived from the NSHE was higher than that from the KYRBS for both sexes with no 95% CI overlapping (Table 2). The differences in the overall obesity prevalence of boys between the data sources in 2018 and 2019 were 4.0%p and 4.7%p, respectively. Similarly, the difference was greater in 2019 among girls—the figures were 5.7%p and 6.3%p, respectively. The annual difference in the overall obesity prevalence of boys and girls based on self-reported data were +0.4%p and +0.1%p. However, the annual differences based on direct measurement were +1.1%p for boys and +0.7%p for girls.

#### 3.1.1. Geographical Distribution of Obesity Prevalence

There were no perfect concordances in geographic distribution regardless of sexes and survey years between the data sources. The Spearman correlation coefficients (*r_s_*) for boys and girls were 0.56 (*p* = 0.02) and 0.53 (*p* = 0.03), respectively, in 2018 and 0.72 (*p* < 0.01) and 0.82 (*p* < 0.01), respectively, in 2019. The ranking discrepancies of regional obesity prevalence for boys were prominent in the US for two consecutive years and was noticeable among girls in 2018 in BS. For boys, the regions with the highest relative ranking were not consistent between the data sources in 2018; the region with the highest rank from the KYRBS was JJ, the provincial island in the south, but that from the NSHE was GW, a province located in the north near the demilitarized zone. Similarly, the regions with the highest rank were not uniform for girls between the data sources in 2019; those from the KYRBS and NSHE were GW and JJ, respectively. However, the region with the highest rank for boys in 2019 and for girls in 2018 was equivalent between the data sources, JJ.

#### 3.1.2. Comparisons of Regional Obesity Prevalence between the KYRBS and NSHE

The numerical agreements of regional adolescent obesity prevalence between the KYRBS and NSHE were evaluated (Figure 3). The levels of regional obesity prevalence estimated using the self-reported data were consistently lower than those derived from direct measurement (dots are located above the 45-degree lines in both graphs) for both sexes. Dots representing US, where the relative ranking discrepancy was prominent among boys for two consecutive years, were found closest to the 45-degree lines. A similar phenomenon was observed among girls in BS in 2018. The dots representing the top three regions with the highest obesity prevalence estimated from the NSHE were mostly found far away from the 45-degree lines for both sexes. In 2018, the regions ranked 1st, 2nd, and 3rd for boys were GW, JJ, and GB, respectively, and for girls were JJ, GW, and JN, respectively. In 2019, those for boys were JJ, GW, and IC, respectively, and for girls were JJ, GB, and GW, respectively. However, the dot representing the obesity prevalence of boys in JJ in 2019 was as closer to the 45-degree line as US.

### 3.2. Adolescent Overweight Prevalence

Like the obesity prevalence, the overall overweight prevalence estimated from directly measured data was higher than that from self-reported data (Table 3). While there was no overlapping of the corresponding CIs in girls between the data sources for two consecutive years, this situation was observed in the 2018 data for boys. The differences in the overall overweight prevalence of boys between the data sources in 2018 and 2019 were 0.9%p and 0.1%p, respectively. Among girls, the differences were greater (1.5%p and 1.3%p, respectively), but were not as prominent as that of obesity prevalence. The annual differences in the overall overweight prevalence of boys and girls based on the KYRBS were +0.7%p and +0.0%p, respectively. Notably, the annual difference for each sex based on the NSHE was decrementing—−0.1%p for boys and −0.2%p for girls. 

#### 3.2.1. Geographical Distribution of Overweight Prevalence

Table 3 represents the geographical distribution of overweight prevalence of adolescents estimated from the 2018–2019 KYRBS and NSHE. In accordance with the obesity prevalence, regional overweight prevalence did not show perfect concordances between the data sources, but there were some differences. First, the negative coefficient indicating the possibly inverse order of relative rankings among boys between the 2018 data sources was not statistically significant (*r_s_* = −0.01; *p* = 0.98). However, no negative figure was detected between the 2019 data sources and the correlation was poorer (*r_s_* = 0.27) and the corresponding *p*-value was insignificant (*p* = 0.29). Although girls showed non-negative Spearman correlation coefficients, the correlations were as poor as the boys’ in 2018 (2018 *r_s_* = 0.15, *p* = 0.56; 2019 *r_s_* = 0.23, *p* = 0.37). The regions with marked ranking discrepancies among boys in 2018 and 2019 were JN and SJ, respectively. Ranking discrepancies for girls were prominent in IC for two consecutive years. The regions with the highest relative ranking differed according to sexes, survey years, and reporting methods. 

#### 3.2.2. Comparisons of Regional Overweight Prevalence between the KYRBS and NSHE

Figure 4 shows simple correlation graphs for regional overweight prevalence of adolescents between self-reported and directly measured data. Compared to the patterns observed in regional obesity prevalence, most of the dots representing overweight prevalence were located closer to the 45-degree lines in both survey years. The discordance in regional overweight prevalence was a little more noticeable among girls than boys, especially in 2018. It was also notable that some scattered dots were located under the 45-degree lines (2018: JN, JB, and SU for boys and JB and GB for girls; 2019: GB, DG, US, SU, GN, JJ, and CB for boys and JJ for girls), which means that the levels of overweight prevalence in the regions represented by these dots were higher when estimated from the KYRBS than from the NSHE. However, the differences detected were not statistically significant. There were some kinds of agreement between the overweight prevalence between the KYRBS and NSHE for boys of US in 2018 and in IC in 2019, as well as girls in JB in 2019. 

## 4. Discussion

In this study, recent Korean adolescent obesity and overweight prevalence were estimated at the regional levels by comparing two nationally representative data sources. BMIs were calculated by applying the newly revised 2017 Growth Chart.

### 4.1. Implications

This study examined the geographical distribution of adolescent obesity and overweight status by determining the relative rankings of each region and investigating the numerical agreements of prevalence between the KYRBS and NSHE. No perfect concordances in the geographic patterns between the data sources were observed according to both sexes and survey years. However, the study results implied some noteworthy facts.

#### 4.1.1. Regional Adolescent Obesity Status

Regardless of data sources used, for both boys and girls, there were no 95% CI overlapping between the least and most obese regions. Obesity prevalence of boys estimated from the 2019 KYRBS ranged from 10.0 (7.2–13.2, SJ) to 21.8 (18.0–25.6, JJ), and that identified from the NSHE ranged from 15.3 (13.3–17.3, GJ) to 23.0 (19.9–26.1, JJ). Similarly, obesity prevalence of girls estimated from the 2019 KRYBS ranged from 4.7 (2.6–6.9, SJ) to 10.6 (7.4–13.8, GW), and that identified from the NSHE ranged from 12.3 (10.1–14.5, SJ) to 19.9 (16.7–23.1, JJ). This result showed that there is some degree of difference in the levels of adolescent obesity across regions. This suggests possible differences in adolescents’ health behaviors by region.

Determining relative rankings of obesity prevalence for each region was also a useful and easy way to identify the area whose prevalence is the highest. The exact match of relative rankings was observed in JJ (1st) for boys in the 2019 and for girls in the 2018 comparisons. The rank differences for boys in 2018 and for girls in 2019 were only one rank away. Some studies focusing on the issue of the area have been completed previously [25,26,27,28], but they were based on the analysis of the KYRBS. The present study results support these study results. Another region with noticeable obesity prevalence was GW. The region showed a moderate level of ranking discrepancy between the 2018 data, but the ranking discrepancy decreased in 2019. This could be interpreted as an improvement in the level of BMI self-perception among adolescents in GW, but it is more likely to indicate that the adolescent obesity problem of the area is getting serious. To the best of our knowledge, there are no studies dealing with the adolescent obesity issue of GW in depth. A study conducted in Japan reported the association between geographical differences in BMI among Japanese school-aged children and geographical differences in day length [29]. Considering the present study results and the geographical location of GW, further studies are required.

Taking the significant and fair to high *r_s_* into consideration, it is suggested that the relative rankings of each region are determined using the two data sources when evaluating regional adolescent obesity levels. However, it should be noted that a large sample size could affect the significance of *p*-values so, caution should be taken if poor reliability (*r_s_* ≤ 0.3 or 0.4) is obtained with a highly significant corresponding *p*-value [30]. Unlike the 2018 comparisons, the *r_s_* obtained in the results from the 2019 data comparisons were higher. In other words, the results indicate that the order of the relative rankings of adolescent obesity prevalence by region showed higher similarities between the data sources in 2019. This may imply that the orders of relative rankings are becoming fixed at certain levels compared to the previous year. If this phenomenon is continuously observed, there is a higher possibility that the ranking discrepancy may become more prominent in the near future. In this study, the 2020 data comparison was not completed since the 2020 NSHE data could not be obtained due to the COVID-19 pandemic. Nonetheless, it is suggested that such comparisons should be continued.

The following should be considered in the examination of relative rankings for adolescent obesity prevalence at the regional levels. First, the same conclusion might not be derived if another method for determining relative rankings, such as the one assigning the same rank to the same prevalence by considering the prevalence only up to the first decimal place or the one returning the average rank to the ranks of the same figures, was used. Hence, specific descriptions about how relative rankings were determined must be provided and followed. Second, the severity of the adolescent obesity status of a region cannot be immediately determined by comparing relative rankings alone. Rather than just investigating *r_s_* and its significance, it is necessary to examine whether the corresponding CIs overlap with that of other regions’ when investigating a region with the highest adolescent obesity prevalence, as in JJ. Most of all, the observed correlations from examining relative rankings between the data sources are not always equivalent to the numerical agreements of prevalence between the data sources.

In addition to determining the level of correlation and its significance using relative rankings, it was also necessary to examine the degree of numerical agreements applying simple correlation graphs, also called scatter diagrams. Although this is not as sensitive as the Bland-Altman plot, it can easily be used in an evaluation of data agreements [31]. The dots representing the levels of adolescent obesity prevalence for each region were all located above the 45-degree lines in the simple correlation graphs. This indicates that the level of numerical agreement of adolescent obesity prevalence of each region estimated using the KYRBS (self-reported) was lower than that derived from the NSHE (directly measured) for two consecutive years regardless of sexes and municipalities. Although this may be the result of measurement errors or variabilities of study participants for each survey, it is more likely to suggest BMI underreporting in the KYRBS in all seventeen municipalities. A previous study which examined the validity of self-reporting using the 2008 KYRBS [32] reported that obese adolescents tended to underreport weight and to overreport height more than non-obese adolescents. Considering this result and the above-mentioned results associated with relative rankings, the numerical discrepancies of regions with higher relative rankings of obesity prevalence derived from the NSHE—especially those represented by dots located far from the 45-degree line in the graphs—should be noted. This could be interpreted as consequences of higher adolescent obesity prevalence but also as a tendency of a low self-perception of obesity. In other words, a BMI underreporting tendency of a region could be easily, simply, and quickly be determined by using both relative rankings and simple correlation graphs. Another noticeable fact from the simple correlation graphs was that the dots representing the regions with prominent ranking discrepancies—boys in US for two consecutive years and girls in BS in 2018—were found closest to the 45-degree lines compared to other dots. This suggests high concordances between adolescent obesity prevalence in these regions, meaning that evaluating relative ranking alone would lead to inaccurate interpretations.

To summarize, relative rankings may be useful together with simple correlation graphs in the evaluation of adolescent obesity at the regional level, but the issue of BMI underreporting should always be acknowledged.

#### 4.1.2. Regional Adolescent Overweight Status

The order of relative rankings did not show any significant similarities between the KYRBS and NSHE as it did for obesity prevalence. In particular, the negative *r_s_* for boys in the 2018 data was unexpected although it showed no significance. In the simple correlation graphs, the dots representing overweight prevalence were located closer to the 45-degree lines, while the dots representing the obesity prevalence of each region were all located above the lines (note the differences in the ranges of the x- and y-axes between the simple correlation graphs for obesity and overweight prevalence). This suggested that there is no significant difference in regional adolescent overweight prevalence between the data sources, meaning that the number of students who accurately report their height and weight in the overweight group is higher.

Moreover, the study results suggested that a further investigation of the regions with prominent ranking discrepancies between the data sources using simple correlation graphs is required regardless of the level and significance of *r_s_*. The regions with prominent ranking discrepancies among boys were JN (rank difference = 15) in 2018 and SJ (rank difference = 16) in 2019. The dot representing JN, marked with one blue asterisk in the 2018 graph, was observed under the 45-degree lines in the simple correlation graphs. This indicated that there is a possibility that the overweight prevalence of JN estimated from the KYRBS was higher than that calculated from the NSHE. Considering the BMI underreporting tendency in the obese group of the area [29], the ranking discrepancy could be caused by the BMI underreporting in the obese group, implying misclassification. On the other hand, the prominent ranking discrepancy of SJ among boys in 2019 (represented by two blue asterisks in the graph) and of IC among girls (marked with one red asterisk) for two consecutive years (rank differences = 14 and 10, respectively) could hardly be explained as the result of BMI underreporting in the obese group as a whole. The dots representing the region indicated probable overweight underreporting in the areas as well. These imply that there is an overall BMI underreporting tendency among male and female students in the areas. Not only is analyzing future trends of BMI underreporting tendencies in the region important but also appropriate educational programs dealing with self-perception of body images and weight management programs are required for these regions. Meanwhile, a different analysis method might be more helpful considering the regional characteristics of SJ. Although the weighted percentages were calculated in the study, it would be extremely meaningful to directly measure the height and weight of all adolescents in the region by designing a cohort group, considering the small number of school-aged children compared to other regions [33]. This could be possible with fiscal support from the local government.

To summarize, it seems relevant that an establishment of prevention and management policies for adolescent overweight problems should be prepared at the central levels, without any regional discrimination. Nonetheless, the local governments and associated educational institutions in the regions with prominent relative ranking discrepancies could propose and implement unique policies considering the regional characteristics analyzed.

### 4.2. Value of Using the Nationally Representative Data for Regional-Level Investigations

In Korea, there are three major types of data sources which collect adolescent height and weight data at national levels, including the KYRBS, NSHE, and the well-known Korea National Health and Nutrition Examination Survey (KNHANES). Many researchers, especially those in clinical fields, prefer the KNHANES since the measurement method is well-standardized, data collection is completed by trained personnel, and clinical information such as laboratory results are also provided. Nonetheless, it may not be useful at regional levels since it was not solely designed for children and adolescents so its sample size for this age group is smaller (approximate 1500 individuals per year) than the KYRBS or NSHE.

Many researchers examining obesity-related problems in adolescence prefer using the KYRBS [34,35,36], since it provides structurally designed questionnaires regarding health behaviors and socioenvironmental factors. However, the most vulnerable feature of the KYRBS is that participants self-report their height and weight [24,32]. Some epidemiological studies have proven the underreporting problems of self-reporting techniques when estimating obesity and overweight status [31,32,37]. However, little research has been completed about how to use self-reported BMI data at regional levels, or whether it is appropriate to use. A study conducted in the United States concluded that state-level obesity estimates based on self-reported data may be misleading [38].

On the other hand, the vulnerability of the NSHE comes from its weakly designed survey questionnaires despite the fact that the direct measurement of height and weight is its strength. Some questions included in the survey lack specificity and some contain ambiguous reference periods. The data collected in this way are useful for health counseling purposes or for determining whether a target person presents any symptoms or diseases. Due to this fatal vulnerability, the number of studies using NSHE data is scarce.

Considering these weaknesses and strengths of the KYRBS and NSHE, it seems meaningful to use both data sources in the examination of adolescent obesity and overweight prevalence at regional levels. This is also supported by the fact that the data collection processes of both surveys are completed in the context of school environments where the fiscal support from taxes is received. The scientific evidence obtained from the analysis could be efficiently used as a foundation to solve related problems at regional levels.

### 4.3. Research Limitations and Significances

The present study has several limitations. First, the validity of self-reported and directly measured height and weight was not examined in detail due to the participant discrepancy. However, the sample selection method for both surveys supposedly ensured its representativeness for the population. Second, the anthropometric measurements were not performed by the same investigators for the NSHE since every school chose its own medical institutions. The calibration or measurement tool types would have produced errors. Additionally, data input errors cannot be fully ignored since school staff is in charge of data input to the NEIS for the measurements of those who are not subject to the specific clinical tests (8th, 9th, 11th, and 12th grades). This is a huge problem not only in the data reliability aspect but also as a factor that could deteriorate education quality because it could be a burden for the school staff. Considering the implications of the NSHE data, it would be necessary to expand targets to all adolescents, to re-examine and revise its survey questionnaires, and to standardize its measuring process more accurately. Third, adolescents outside schools could not be included in both the KYRBS and NSHE. Considering these limitations, a long-term analysis comparing the two nationally representative data in the aspect of adolescent obesity and overweight status at the regional level should be continued.

The strengths of this study lie in the following. First, this study suggested a way to utilize two nationally representative data, especially the NSHE whose use for research purposes was relatively low even though the height and weight of adolescents are directly measured. Second, this study suggested an easy and simple way to grasp ideas about current adolescent obesity and overweight status at the regional level. These methods would be useful for those with limited abilities to handle statistical packages designed for professional purposes or those with a tremendous amount of work to complete—such as many public servants. Third, this study proposed that it is crucial to categorize participants into an obese group and an overweight group distinctively when dealing with the obesity problem. Fourth, the study results also highlighted the importance of looking into the regions with prominent ranking discrepancies in overweight prevalence between the data sources even if the level of *r_s_* is poor and the corresponding *p*-value is insignificant.

The planning and execution of health-policies associated with the prevention and management of obesity in adolescence are mainly developed by the central government, represented by the Ministry of Health and Welfare. According to the first Master Plan for Student Health Promotion (2019~2023), the local education offices should prepare community-appropriate strategies by considering the conditions and characteristics of each region, while schools should manage local communities and related institutions [39]. It seems adolescent obesity issue is entrusted only to the local system and the central government put little effort to consider about the relevance and liability of applying the policies to each region [40]. In order to prevent and manage adolescent obesity in the aspect of health inequality between regions, the followings should be implemented. First, a development of valid adolescent health monitoring system should be prepared. Unlike the KYRBS which is conducted and managed by the Centers for Disease Control Agency, the Ministry of Education deals with the NSHE. Taking the strength and weakness of each data sources into consideration, both organizations should deeply reconsider about the data collection process and revise if necessary. By doing so, sufficient and valid evidence would be generated at the national and regional levels. Second, establishment of a governance to solve adolescent obesity problem is necessary. Centered by the Ministry of Health and Welfare and the Ministry of Education, health-related sectors, including the Ministry of Environment, local education offices, and school districts, should cooperate to resolve the problem [39]. The central government could allocate more grants to the most obese regions so that establishment of relevant related policies to help produce positive results, including the enhancement of health-related behaviors and the reduction in diseases, disabilities, or unexpected deaths in adulthood can take a place. Some effective policy options in reducing adolescent obesity suggested by an Australian study included nutrition education, physical education, and parental involvement in such activities [41]. It would be also possible for the government to develop daily physical activity guidelines based on the obesity level of each region and to incentivize schools which meet the proposed guidelines and show improvements [42].

## 5. Conclusions

This study showed that two different nationally representative data could be used in the examination of the adolescent obesity burden at the regional level. The importance of analyzing the geographical distribution of adolescent obesity and overweight status is emphasized by the study. The study results are expected to be used as scientific evidence in the establishment of adolescent obesity prevention and management policies and their execution both at the central and regional levels.

## Figures and Tables

**Figure 1 healthcare-09-01751-f001:**
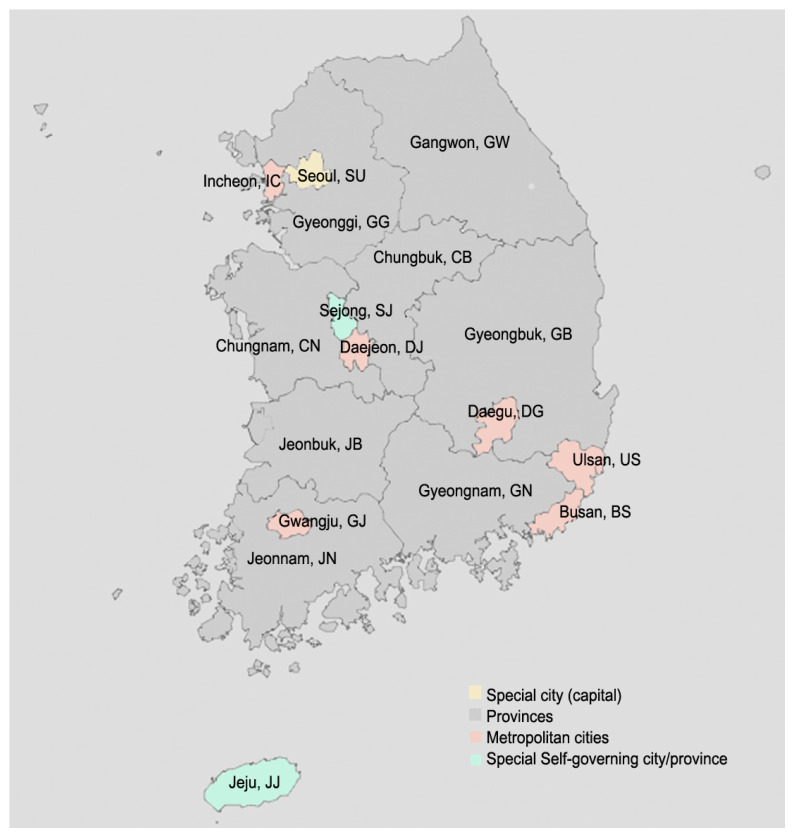
Regional divisions of South Korea. Seoul (SU) is the capital city of the country and colored in yellow. Those colored in pink are metropolitan cities. Sejong (SJ), the self-governing city, and Jeju (JJ), the self-governing province are colored in light green. Provinces are colored in gray.

**Figure 2 healthcare-09-01751-f002:**
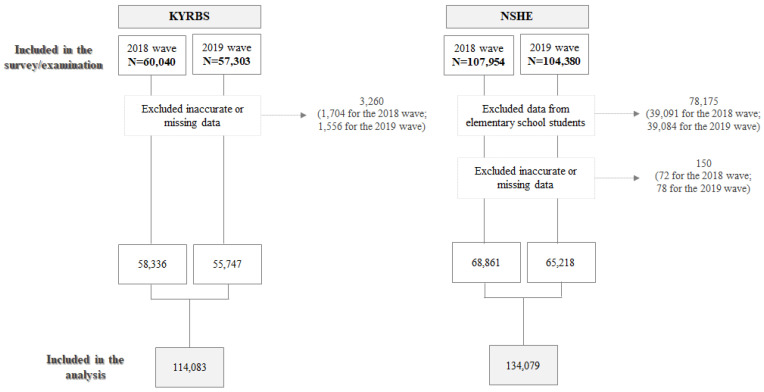
Participants included in the analysis.

**Figure 3 healthcare-09-01751-f003:**
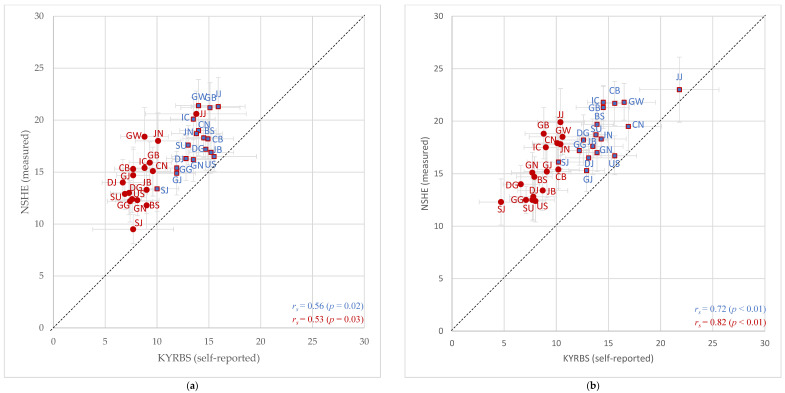
Simple correlation graphs for regional obesity prevalence estimated from the KYRBS (self-reported) and NSHE (directly measured). (**a**) Comparisons of 2018 prevalence between the data sources; (**b**) comparisons of 2019 prevalence between the data sources. Dots represent weighted percentage and gray lines indicate 95% confidence intervals. Values for boys are presented with blue (square) dots and those for girls are presented with red (circle) dots. Abbreviations: SU, Seoul; BS, Busan; DG, Daegu; IC, Incheon; GJ, Gwangju; DJ, Daejeon; US, Ulsan; SJ, Sejong; GG, Gyeonggi; GW, Gangwon; CB, Chungbuk; CN, Chungnam; JB, Jeonbuk; JN, Jeonnam; GB, Gyeongbuk; GN, Gyeongnam; JJ, Jeju.

**Figure 4 healthcare-09-01751-f004:**
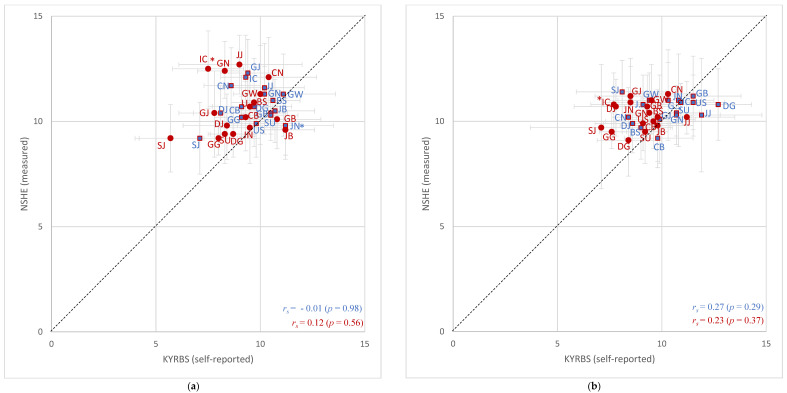
Simple correlation graphs for regional overweight prevalence estimated from the KYRBS (self-reported) and NSHE (directly measured). (**a**) Comparisons of 2018 prevalence between the data sources; (**b**) comparisons of 2019 prevalence between the data sources. Dots represent weighted percentage and gray lines indicate 95% confidence intervals. Values for boys are presented with blue (square) dots and those for girls are presented with red (circle) dots. Regions with prominent ranking discrepancies are marked with asterisk(s) and detailed descriptions are found in Section 4.1.2. Abbreviations: SU, Seoul; BS, Busan; DG, Daegu; IC, Incheon; GJ, Gwangju; DJ, Daejeon; US, Ulsan; SJ, Sejong; GG, Gyeonggi; GW, Gangwon; CB, Chungbuk; CN, Chungnam; JB, Jeonbuk; JN, Jeonnam; GB, Gyeongbuk; GN, Gyeongnam; JJ, Jeju.

**Table 1 healthcare-09-01751-t001:** Number and mean age of study participants of the KYRBS and NSHE (2018–2019).

Category	Subgroup	2018				2019			
		KYRBS (N = 58,336)		NSHE (N = 68,861)		KYRBS (N = 55,747)		NSHE (N = 65,218)	
		Boys	Girls	Boys	Girls	Boys	Girls	Boys	Girls
N	Total	29,613	28,723	34,928	33,933	29,058	26,689	32,958	32,260
Region	SU	4276	4271	2931	2727	4078	4025	2620	2578
	BS	1822	1890	1930	1892	1821	1601	1848	1827
	DG	1431	1553	2054	1934	1569	1322	1981	1825
	IC	1759	1539	1736	1807	1776	1389	1654	1759
	GJ	1116	1261	1349	1292	1084	1158	1379	1377
	DJ	1192	1025	1975	1913	1185	938	1850	1764
	US	1011	881	1988	1993	1035	746	1733	1849
	SJ	472	452	1072	1054	489	442	1056	1113
	GG	6201	6246	4956	5047	6105	5942	4805	4665
	GW	1134	1056	1734	1568	1090	917	1606	1481
	CB	1180	1122	2279	1727	1116	1025	2118	2000
	CN	1328	1190	1852	1624	1261	1229	1772	1452
	JB	1277	1137	1714	2124	1193	1189	1627	1614
	JN	1145	1235	1508	1507	1325	1154	1348	1532
	GB	1483	1476	2306	2121	1440	1476	2198	1982
	GN	1924	1851	2258	2109	1820	1572	2111	1993
	JJ	862	538	1286	1494	671	564	1252	1449
Age, mean ± SE	Total	15.2 ± 0.05	15.1 ± 0.05	15.0 ± 0.04	15.0 ± 0.04	15.1 ± 0.05	15.1 ± 0.05	14.9 ± 0.04	14.9 ± 0.04

Abbreviations: KYRBS, Korea Youth Risk Behavior Web-based Survey; NSHE, National School Health Exam; N, number; SE, Standard error; SU, Seoul; BS, Busan; DG, Daegu; IC, Incheon; GJ, Gwangju; DJ, Daejeon; US, Ulsan; SJ, Sejong; GG, Gyeonggi; GW, Gangwon; CB, Chungbuk; CN, Chungnam; JB, Jeonbuk; JN, Jeonnam; GB, Gyeongbuk; GN, Gyeongnam; JJ, Jeju.

**Table 2 healthcare-09-01751-t002:** Regional obesity prevalence and relative rankings from the KYRBS and NSHE (2018–2019).

Year		2018					2019				
Sex	Region	KYRBS		NSHE		Rank Difference	KYRBS		NSHE		Rank Difference
		w% (95% CI)	Rank	w% (95% CI)	Rank		w% (95% CI)	Rank	w% (95% CI)	Rank	
Boys	Overall	13.4 (12.9–13.8)	-	17.4 (16.8–17.9)	-	-	13.8 (13.3–14.3)	-	18.5 (17.9–19.1)	-	-
	SU	13.0 (12.1–14.0)	13	17.6 (15.7–19.5)	9	4	13.8 (12.5–15.1)	11	18.7 (16.8–20.6)	8	3
	BS	14.5 (12.8–16.2)	7	18.3 (16.4–20.1)	7	0	13.9 (12.4–15.5)	10	19.7 (17.6–21.7)	6	4
	DG	14.7 (13.2–16.2)	6	17.2 (15.2–19.3)	10	4	12.6 (10.3–14.9)	15	18.2 (15.8–20.6)	10	5
	IC	13.5 (11.3–15.6)	12	20.1 (18.1–22.2)	4	8	14.5 (13.0–15.9)	6	21.8 (20.1–23.4)	3	3
	GJ	11.9 (10.0–13.7)	16	14.9 (12.8–17.0)	16	0	12.9 (11.2–14.6)	14	15.3 (13.3–17.3)	17	3
	DJ	12.8 (10.9–14.7)	14	16.3 (14.2–18.5)	13	1	13.1 (10.9–15.3)	13	16.5 (14.3–18.7)	15	2
	US	15.5 (11.4–19.6)	2	16.5 (15.1–17.9)	12	10	15.6 (13.6–17.7)	5	16.7 (14.9–18.6)	14	9
	SJ	10.0 (7.8–12.1)	17	13.4 (11.2–15.6)	17	0	10.2 (7.2–13.2)	17	16.1 (13.4–18.7)	16	1
	GG	11.9 (11.0–12.8)	15	15.4 (14.2–16.6)	15	0	12.2 (11.2–13.2)	16	17.2 (15.8–18.6)	12	4
	GW	14.0 (11.8–16.1)	8	21.4 (18.9–23.9)	1	7	16.5 (13.6–19.5)	3	21.8 (20.1–23.6)	2	1
	CB	14.9 (12.5–17.4)	5	18.2 (16.3–20.1)	8	3	15.6 (13.7–17.5)	4	21.7 (19.6–23.8)	4	0
	CN	14.0 (12.2–15.7)	9	19.0 (16.6–21.3)	5	4	16.9 (13.8–20.1)	2	19.5 (17.2–21.7)	7	5
	JB	15.2 (13.1–17.4)	3	16.9 (14.9–18.9)	11	8	13.5 (10.6–16.4)	12	17.6 (14.9–20.2)	11	1
	JN	13.8 (11.4–16.3)	10	18.7 (15.9–21.5)	6	4	14.3 (12.0–16.7)	8	18.3 (16.5–20.2)	9	1
	GB	15.1 (12.3–18.0)	4	21.2 (18.8–23.6)	3	1	14.5 (11.9–17.0)	7	21.3 (19.3–23.3)	5	2
	GN	13.5 (11.5–15.6)	11	16.2 (14.1–18.2)	14	3	13.9 (12.3–15.6)	9	17.0 (15.2–18.8)	13	4
	JJ	15.9 (13.3–18.5)	1	21.3 (18.6–24.1)	2	1	21.8 (18.0–25.6)	1	23.0 (19.9–26.1)	1	0
	*r_s_*	0.56 (*p* = 0.02)					0.72 (*p* < 0.01)				
Girls	Overall	8.0 (7.6–8.5)	-	13.7 (13.1–14.2)	-	-	8.1 (7.7–8.6)	-	14.4 (13.8–15.0)	-	
	SU	6.9 (5.8–8.1)	16	12.9 (11.4–14.5)	12	4	8.0 (7.0–9.0)	10	12.4 (10.4–14.4)	16	6
	BS	9.0 (7.9–10.1)	5	11.8 (9.5–14.0)	16	11	7.9 (6.2–9.6)	11	14.7 (12.9–16.6)	10	1
	DG	7.3 (5.5–9.1)	15	13.0 (10.9–15.1)	11	4	6.6 (4.9–8.4)	16	14.0 (12.2–15.8)	11	5
	IC	8.8 (6.7–10.8)	7	15.4 (12.6–18.2)	5	2	9.0 (6.9–11.2)	7	17.5 (15.3–19.7)	6	1
	GJ	7.7 (6.1–9.3)	12	14.7 (12.0–17.4)	8	4	9.1 (6.9–11.4)	6	15.2 (12.7–17.8)	8	2
	DJ	6.7 (4.7–8.7)	17	14.0 (11.8–16.2)	9	8	7.8 (5.5–10.0)	12	12.8 (10.6–15.1)	13	1
	US	7.4 (5.2–9.6)	14	12.2 (10.2–14.2)	15	1	7.7 (5.5–9.8)	14	12.5 (10.5–14.5)	14	0
	SJ	7.7 (3.8–11.6)	11	9.5 (8.1–10.9)	17	6	4.7 (2.6–6.9)	17	12.3 (10.1–14.5)	17	0
	GG	7.6 (6.7–8.5)	13	12.4 (11.1–13.7)	13	0	7.1 (6.3–7.9)	15	12.5 (11.2–13.8)	15	0
	GW	8.8 (6.5–11.1)	8	18.4 (15.6–21.2)	2	6	10.6 (7.4–13.8)	1	18.5 (16.8–20.3)	3	2
	CB	7.7 (5.9–9.5)	10	15.3 (13.5–17.2)	6	4	10.2 (8.2–12.2)	4	15.4 (13.6–17.3)	7	3
	CN	9.6 (7.9–11.3)	3	15.1 (12.7–17.4)	7	4	10.1 (7.7–12.5)	5	17.9 (15.3–20.4)	4	1
	JB	9.0 (7.3–10.6)	6	13.3 (11.0–15.7)	10	4	8.7 (6.3–11.1)	9	13.4 (11.5–15.4)	12	3
	JN	10.1 (7.9–12.3)	2	18.0 (15.3–20.7)	3	1	10.4 (7.7–13.1)	3	17.8 (15.5–20.1)	5	2
	GB	9.3 (7.3–11.4)	4	15.9 (13.8–17.9)	4	0	8.8 (7.0–10.6)	8	18.8 (16.4–21.3)	2	6
	GN	8.1 (6.0–10.2)	9	12.3 (10.8–13.8)	14	5	7.7 (5.7–9.6)	13	15.1 (13.1–17.0)	9	4
	JJ	13.8 (8.9–18.6)	1	20.6 (18.5–22.8)	1	0	10.4 (8.2–12.7)	2	19.9 (16.7–23.1)	1	1
	*r_s_*	0.53 (*p* = 0.03)					0.82 (*p* < 0.01)				

Abbreviations: KYRBS, Korea Youth Risk Behavior Web-based Survey; NSHE, National School Health Exam; w%, weighted percentage; CI, confidence interval; SU, Seoul; BS, Busan; DG, Daegu; IC, Incheon; GJ, Gwangju; DJ, Daejeon; US, Ulsan; SJ, Sejong; GG, Gyeonggi; GW, Gangwon; CB, Chungbuk; CN, Chungnam; JB, Jeonbuk; JN, Jeonnam; GB, Gyeongbuk; GN, Gyeongnam; JJ, Jeju; *r_s_*, Spearman correlation coefficient. BMI ≥ 95th percentile was defined as obese. Based on the weighted percentages (up to three decimal points were considered), relative rankings (rank) are designated with lower numbers for the most obese regions and with higher numbers for the least obese regions.

**Table 3 healthcare-09-01751-t003:** Regional overweight prevalence and relative rankings from the KYRBS and NSHE (2018–2019).

Year		2018					2019				
Sex	Region	KYRBS		NSHE		Rank Difference	KYRBS		NSHE		Rank Difference
		w% (95% CI)	Rank	w% (95% CI)	Rank		w% (95% CI)	Rank	w% (95% CI)	Rank	
Boys	Overall	9.8 (9.4–10.1)	-	10.7 (10.3–11.1)		-	10.5 (10.1–10.9)	-	10.6 (10.2–11.0)	-	-
	SU	10.5 (9.7–11.3)	5	10.4 (9.4–11.5)	11	6	10.7 (9.5–11.8)	8	10.4 (9.3–11.5)	10	2
	BS	10.6 (9.4–11.9)	4	11.0 (9.6–12.4)	7	3	9.0 (7.6–10.5)	14	9.7 (8.2–11.2)	16	2
	DG	9.7 (8.6–10.8)	10	10.7 (9.3–12.1)	8	2	12.7 (11.0–14.3)	1	10.8 (9.1–12.5)	8	7
	IC	9.3 (8.1–10.5)	12	12.1 (10.1–14.1)	2	10	10.8 (9.1–12.5)	6	11.0 (8.8–13.2)	4	2
	GJ	9.4 (8.1–10.7)	11	12.3 (10.7–13.9)	1	10	9.9 (8.1–11.6)	10	10.1 (8.0–12.2)	14	4
	DJ	8.1 (6.8–9.5)	16	10.4 (9.1–11.7)	12	4	8.6 (6.8–10.4)	15	9.9 (8.5–11.4)	15	0
	US	9.8 (8.0–11.5)	9	9.9 (8.3–11.5)	15	6	11.5 (9.3–13.8)	3	11.2 (9.3–13.1)	2	1
	SJ	7.1 (4.2–10.0)	17	9.2 (7.5–10.9)	17	0	8.1 (5.9–10.2)	17	11.4 (9.9–12.9)	1	16
	GG	9.1 (8.3–9.8)	14	10.2 (9.3–11.2)	14	0	10.9 (10.1–11.8)	5	10.9 (10.0–11.8)	7	2
	GW	11.1 (8.7–13.6)	2	11.3 (9.3–13.2)	6	4	9.4 (7.5–11.3)	12	11.0 (9.6–12.3)	5	7
	CB	9.1 (7.7–10.6)	13	10.7 (9.2–12.2)	9	4	9.8 (8.2–11.5)	11	9.2 (7.8–10.6)	17	6
	CN	8.6 (7.3–10.0)	15	11.7 (9.9–13.5)	3	12	8.4 (7.6–9.3)	16	10.2 (8.7–11.6)	13	3
	JB	10.7 (8.8–12.6)	3	10.5 (9.0–12.0)	10	7	9.1 (6.6–11.7)	13	10.8 (9.3–12.2)	9	4
	JN	11.2 (8.9–13.5)	1	9.8 (8.4–11.3)	16	15	10.3 (8.6–11.9)	9	11.0 (9.2–12.8)	3	6
	GB	10.5 (9.2–11.8)	6	10.3 (9.1–11.6)	13	7	11.5 (9.9–13.0)	4	10.9 (9.6–12.2)	6	2
	GN	10.2 (8.7–11.7)	7	11.3 (9.9–12.6)	5	2	10.7 (8.6–12.8)	7	10.3 (8.9–11.7)	12	5
	JJ	10.2 (8.2–12.1)	8	11.6 (9.4–13.7)	4	4	11.9 (9.0–14.8)	2	10.3 (7.6–13.0)	11	0
	*r_s_*	−0.01 (*p* = 0.98)					0.27 (*p* = 0.29)				
Girls	Overall	8.7 (8.3–9.1)	-	10.2 (9.8–10.6)	-		8.7 (8.4–9.1)	-	10.0 (9.6–10.4)	-	-
	SU	8.3 (7.5–9.1)	12	9.4 (8.0–10.7)	15	3	9.2 (8.1–10.3)	9	9.5 (8.0–11.0)	16	7
	BS	9.7 (8.6–10.8)	5	10.9 (8.8–13.0)	6	1	9.8 (8.2–11.4)	3	10.2 (8.9–11.5)	9	6
	DG	8.7 (7.6–9.8)	10	9.4 (8.3–10.6)	14	4	8.4 (6.9–10.0)	13	9.1 (7.4–10.7)	17	4
	IC	7.5 (5.8–9.2)	16	12.5 (10.6–14.3)	2	14	7.7 (6.6–8.9)	15	10.8 (9.3–12.3)	5	10
	GJ	7.8 (6.1–9.5)	15	10.4 (8.3–12.4)	8	7	8.5 (6.8–10.2)	12	11.2 (9.3–13.0)	2	10
	DJ	8.4 (6.3–10.6)	11	9.8 (8.2–11.3)	11	0	7.8 (5.8–9.7)	14	10.7 (9.3–12.0)	7	7
	US	9.5 (7.5–11.5)	7	10.7 (9.3–12.1)	7	0	9.1 (7.2–11.0)	10	9.9 (8.6–11.1)	12	2
	SJ	5.7 (4.0–7.3)	17	9.2 (7.6–10.8)	16	1	7.1 (3.7–10.4)	17	9.7 (6.8–12.7)	14	3
	GG	8.0 (7.1–8.9)	14	9.2 (8.4–9.9)	17	3	7.6 (7.0–8.3)	16	9.5 (8.7–10.4)	15	1
	GW	10.0 (8.7–11.2)	4	11.3 (8.9–13.6)	5	1	9.5 (7.9–11.2)	6	11.0 (9.4–12.7)	3	3
	CB	9.3 (7.8–10.8)	8	10.2 (8.6–11.9)	9	1	9.6 (7.3–11.9)	5	10.0 (8.7–11.2)	11	6
	CN	10.4 (8.1–12.7)	3	12.1 (10.2–14.0)	4	1	10.3 (8.7–11.9)	2	11.3 (9.1–13.4)	1	1
	JB	11.2 (9.9–12.6)	1	9.6 (8.2–11.1)	13	12	9.8 (8.0–11.5)	4	9.8 (8.1–11.5)	13	9
	JN	9.5 (8.0–11.0)	6	9.7 (8.0–11.4)	12	6	8.5 (7.2–9.8)	11	10.9 (9.2–12.6)	4	7
	GB	10.8 (8.7–12.9)	2	10.1 (8.7–11.5)	10	8	9.3 (8.2–10.4)	8	10.7 (9.2–12.2)	6	2
	GN	8.3 (7.0–9.5)	13	12.4 (11.1–13.8)	3	10	9.4 (8.1–10.7)	7	10.4 (9.4–11.4)	8	1
	JJ	9.0 (6.1–12.0)	9	12.7 (11.3–14.1)	1	8	11.2 (8.8–13.5)	1	10.2 (8.4–11.9)	9	8
	*r_s_*	0.12 (*p* = 0.56)					0.23 (*p* = 0.37)				

Abbreviations: KYRBS, Korea Youth Risk Behavior Web-based Survey; NSHE, National School Health Exam; w%, weighted percentage; CI, confidence interval; SU, Seoul; BS, Busan; DG, Daegu; IC, Incheon; GJ, Gwangju; DJ, Daejeon; US, Ulsan; SJ, Sejong; GG, Gyeonggi; GW, Gangwon; CB, Chungbuk; CN, Chungnam; JB, Jeonbuk; JN, Jeonnam; GB, Gyeongbuk; GN, Gyeongnam; JJ, Jeju; *r_s_*, Spearman correlation coefficient. BMI ≥ 95th percentile was defined as obese. Based on the weighted percentages (up to three decimal points were considered), relative rankings (rank) are designated from lower numbers for the most obese regions and with higher numbers for the least obese regions.

## Data Availability

The KYRBS data are available from the KDCA (URL: http://www.kdca.go.kr/yhs/ (accessed on 23 November 2020)). The NSHE data are available from the Ministry of Education (URL: https://www.schoolhealth.kr/web/srs/selectSchoolOverview.do?sMenuId=0100008600 (accessed on 27 October 2020)).

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
