# Peer review of "Regional Adolescent Obesity and Overweight Status in Korea from 2018–2019: Comparison between Two Data Sources"

_healthcare, 2021, doi:10.3390/healthcare9121751_

Round 1

Reviewer 1 Report

The authors examined regional adolescent obesity and overweight status by comparing two data sources, one of which got their latest height and weight by self-report and the other got them by directly measurement. They demonstrate the discrepancy and the correlations in prevalence of obese and overweight when divided by regions. It is interesting that the obese adolescence people tended to underreport their weight, but the overweight adolescence people little underreport it. However, I am sorry, but I don’t think the analyses and conclusions have scientific importance. Although the correlations in prevalence of obese were statistically significant, the regional differences vary between 2018 and 2019.  The authors insist this analysis suggests how we can investigate the geographical distribution of adolescent obesity status based on two nationally representative data sources. However, it does not demonstrate the specific method to the regional public health. The only fact that we get from this manuscript is that height and weight of adolescence are better to actually measure than just ask, and unfortunately requires no research paper to be demonstrated because it's not beyond the bounds of common sense.

Author Response

  • Thank you for your valuable comments and I am sorry that this study did not meet your expectations. I totally understand your suspicion regarding the scientific importance of this study. Nevertheless, we do believe that research on how to make use of available data should not be neglected and is as important as discovering new scientific knowledge. As you pointed out, direct measurement of height and weight of adolescence is better, and it is not beyond the bounds of common sense. However, the KYRBS collects height and weight by self-reporting so that regional obesity estimation may be misleading considering previous studies conducted abroad. On the other hand, the NSHE directly measures height and weight, but its vulnerability comes from its weakly designed survey questionnaires. We, the authors of this study, wanted to prove the followings; 1) the value of examining obesity and overweight status of adolescents at the regional levels considering health inequality issue, which is getting serious in Korea, and 2) reconsideration of the quality of the available data in order to prove what to make out of it. We’ve made some improvements based on the reviewers’ comments. I would appreciate if you could re-review the revised version if you don’t mind. Thank you.

Reviewer 2 Report

  • Line 11: Please provide a background to make a reader understand the reasons of conducting this study.
  • Line 11-13: The study design should be clarified here. For example, "a cross sectional population-based study using data sources....etc". The date when a study takes place and the age of participants should also be clearly mentioned.
  • Line 18-21: Significant P-value is necessary to display.
  • Line 26-27: Please add “schoolchildren” and “Korea” to the list.
  • Line 30-32: What are the prevalence and risk factors for childhood obesity in Korea?
  • Line 34,41,43,56: Please define here. Is it cross-sectional or longitudinal study? Also make sure to check this throughout the paper.
  • Line 37: “macroscopic approach”. Meaning unclear. Suggest using another term; e.g., “prevention approach”.
  • Line 40-43: Readers need to understand what biological/non-biological factors mean. Please clarify.
  • Line 85-96: Whilst the paper provides an interesting local insight, it may benefit from some broader international context to appeal to a geographically wider readership. For example it would have been useful to have drawn reference to some of the other policy work from Australia, UK and US (please refer to these articles: Children (Basel). 2018 Jan 29;5(2):18.doi 3390/children5020018; Int J Environ Res Public Health. 2020, 17(22):8405), and perhaps to the WHO school policy framework?
  • Figure 1 should be clear and large enough to represent clearly all regional divisions of South Korea.
  • Line 107-116: Lack of sufficient novelty is clear. The significance should be clearly emphasized in the paper. Why this study is important?
  • Line 11-120: Please refer to my comment in abstract. The study design should be cleanly mentioned.
  • Inclusion and exclusion criteria should be reported. Also a figure showing the final participants is needed (Line 140-142, Line 156-158). How authors deal with missing value?
  • Line 134-136, Line 152-154: Data collection procedure should be reported in much more details.
  • Line 171-172: Why Pearson correlation coefficients was not perform? Did authors determine whether a data set is modeled for normal distribution? Why t-test was not used?
  • Line 174: Please provide more details about decimal points.
  • Line 175: This version is old. Why authors not use current versions (25.0/26.0).
  • The discussion should be supported by recent studies from international contexts. Please refer to my comment in introduction. Authors should also view on the following questions: (1) What are the risk factors associated with overweight and obesity in Korea? (2) How can childhood obesity be prevented from a policy level? (3) How can the government help with childhood obesity?  
  • Some references are old (Ref# 4,5,8)- Please update.

Author Response

Abstract

Line 11: Please provide a background to make a reader understand the reasons of conducting this study.

  • Thank you for your time and many valuable comments. We’ve added a statement related to the background of this study in the abstract.

Difference in the regional adolescent obesity level may cause a notable health inequality between regions since it significantly affects adulthood health status.

Line 11-13: The study design should be clarified here. For example, "a cross sectional population-based study using data sources....etc". The date when a study takes place and the age of participants should also be clearly mentioned.

  • It was revised according to your suggestion.

This study examined 2018 and 2019 regional obesity and overweight status of adolescents aged 12 to 18 by comparing two cross sectional population-based data sources, the Korea Youth Risk Behavior Web-based Survey (KYRBS) and the National Student Health Examination (NSHE).

Line 18-21: Significant P-value is necessary to display.

  • It was revised accordingly.

For adolescent obesity status, rs levels were fair to good with p-values less than 0.05, but the correlation graphs indicated body mass index (BMI) underreporting in the KYRBS.

Line 26-27: Please add “schoolchildren” and “Korea” to the list.

  • “schoolchildren” and “Korea” were added as the keywords.

Keywords: adolescent; geographical location; Korea; obesity; overweight; regional health planning; schoolchildren

Introduction

Line 30-32: What are the prevalence and risk factors for childhood obesity in Korea?

  • The recent obesity and overweight prevalence change in Korea were added in the introduction section.

Similarly, recent obesity and overweight trends in Korea are also increasing from 15.3% in 2007 to 23.7% in 2017 [2].

  • We’ve included some of the known risk factors associated with adolescence obesity in Korea in the introduction section as well. Please check.

Adolescence obesity in Korea is known to be associated with many risk factors including low level of fruit and vegetable intake, consumption of ramen, poor sleep hygiene, physical inactivity, and depression [3].

Line 34,41,43,56: Please define here. Is it cross-sectional or longitudinal study? Also make sure to check this throughout the paper.

  • The study designs were indicated as your suggestion.

Line 37: “macroscopic approach”. Meaning unclear. Suggest using another term; e.g., “prevention approach”.

  • “prevention approach” was used as your suggestion. This statement was moved to the discussion section for clarity. Thank you.

Line 40-43: Readers need to understand what biological/non-biological factors mean. Please clarify.

  • “biological/non-biological” was changed to “genetic/non-genetic” for clarification. Thank you for pointing this out.

Line 85-96: Whilst the paper provides an interesting local insight, it may benefit from some broader international context to appeal to a geographically wider readership. For example it would have been useful to have drawn reference to some of the other policy work from Australia, UK and US (please refer to these articles: Children (Basel). 2018 Jan 29;5(2):18.doi 3390/children5020018; Int J Environ Res Public Health. 2020, 17(22):8405), and perhaps to the WHO school policy framework?

  • Thank you for helpful references. I’ve used those in the discussion section. Thanks a lot.

Figure 1 should be clear and large enough to represent clearly all regional divisions of South Korea.

  • We’ve changed the type of font of the figure and made it larger. Please check.

Line 107-116: Lack of sufficient novelty is clear. The significance should be clearly emphasized in the paper. Why this study is important?

  • Despite the fact that the NSHE measure height and weight directly, its vulnerability comes from poorly designed questionnaires. Since the Ministry of Education which has control over school districts is in charge of the data collection, political and academic usefulness of this data was questionable. On the other hands, the Centers for Disease Control Agency is in charge of the KYRBS. Although it does not measure students’ height and weight directly, a lot of reports associated with adolescent obesity are based on this data. The authors of this study were curious about the value of the available NSHE data and about what to make out of it. Since adolescence obesity issue is definitely related to health issues in adulthood, we strongly believe that the results of this study could contribute to resolve health inequality issue between regions. We expect the study results can be used as a basis for the establishment of related policies.

Materials and methods

Line 11-120: Please refer to my comment in abstract. The study design should be cleanly mentioned.

  • The study designs were indicated as your suggestion.

Inclusion and exclusion criteria should be reported. Also a figure showing the final participants is needed (Line 140-142, Line 156-158). How authors deal with missing value?

  • I am sorry about the inconvenience. As described in the methods, the number of students who participated in the 2018 wave was 60040, and 57303 students participated in the 2019 wave for the KYRBS. For the NSHE, the number of students who checked physical developmental status in the 2018 wave was 107954 and 104380 students in the 2019 wave. Data from elementary school students were excluded from the NSHE (39091 for the 2018 wave and 39084 for the 2019 wave) as described in the manuscript. In addition, the number of students excluded from the KYRBS analysis due to inaccuracies in age and weight/height is 3260 (1704 for the 2018 wave; 1556 for the 2019 wave) and that of the NSHE is 150 (72 for the 2018 wave; 78 for the 2019 wave). Therefore, as described in the results, the total number of the KYRBS and NSHE participants included in the analysis were 114083 (58336 for the 2018 wave; 55747 for the 2019 wave) and 134079 (68861 for the 2018 wave; 65218 for the 2019 wave), respectively. As you recommendation a figure (figure 2) showing the final participants was additionally provided. Please check. (Originally indicated figures 2 and 3 were revised to figures 3 and 4 accordingly.)

Line 134-136, Line 152-154: Data collection procedure should be reported in much more details.

  • I do agree with your suggestion and there is some necessity to provide information associated with data collection procedure. However, since this study is dealing with two different data sources and the results of the data is presented according to regions, sexes, and years, a lot of information should be given in the manuscript while there is a word restriction. We tried our best to summarize key points regarding the data collection procedure. I hope you can understand our situation. The followings are detailed descriptions regarding data collection procedure for your understanding. Thank you.

KYRSB: KYRBS has been conducted every year since 2005 and is in charge of Korea Centers for Disease Agency (KCDA). KYRBS was designed to protect students’ privacy and voluntary participation. The trained teacher distributes an information sheet to each student and explains the purpose and participation procedure of the survey using instructional materials in a school computer laboratory where internet access is available. Students log in using the certificate number printed on the information sheet and check online informed consent. Then students complete the anonymous self-administered web-based questionnaire during one class period. It takes about 40 min for students to set up the survey and complete the questionnaires. The KCDA Institutional Review Board has approved the protocols for KYRBS [Kim, Y., et al. Data resource profile: The Korea Youth Risk Behavior Web-based Survey (KYRBS). Int J Epidemiol 2016, 45(4); 1076-1076e]

NSHE: For NSHE, a sufficient number of participants were enrolled every year. However, there were limitations, such as measurement errors and exclusion of children and adolescents not attending school. … NSHE aims to generate baseline data related to health issues of students in elementary, middle, and high schools. The students are assessed annually under the supervision of the Ministry of Education. A two-stage, stratified, cluster sampling method was used in the NSHE to select a representative sample of students in Korea. Recently, more than 80,000 students aged 6 to 18 years participated in the NSHE. [Kim, J.H., Moon, J.S. Secular trends in pediatric overweight and obesity in Korea. J Obes Metab Syndr 2020, 29(1), 12-17.]

Line 171-172: Why Pearson correlation coefficients was not perform? Did authors determine whether a data set is modeled for normal distribution? Why t-test was not used?

  • The study analysis focused on the relative rankings instead of numerical comparisons between the obesity and overweight prevalence since methods used to collect height and weight data were different between the KYRBS (self-reported) and the NSHE (directly measured). We wanted to compare “regional distribution” of obesity status by comparing relative rankings, therefore spearman correlation was performed.

Line 174: Please provide more details about decimal points.

  • I am going to take figures in the Table 3 to explain. For example, Both GN and JJ present overweight prevalence of 10.2 but GN is ranked in the 7th place and JJ is ranked in the 8th Although in the table the prevalence was presented up to one decimal point due to space restriction, the relative ranking was determined considering overweight prevalence up to three decimal points. I hope this can help your understanding.

Line 175: This version is old. Why authors not use current versions (25.0/26.0).

  • We are aware of that the SPSS version we have is outdated. We’ve confirmed the analysis results with the updated version, 25.0. We left the description in the statistical analysis was as it was since the original analysis was completed with the version. We will make sure to update the software. Thank you.

Discussion

The discussion should be supported by recent studies from international contexts. Please refer to my comment in introduction. Authors should also view on the following questions: (1) What are the risk factors associated with overweight and obesity in Korea? (2) How can childhood obesity be prevented from a policy level? (3) How can the government help with childhood obesity? 

  • (1) the risk factors associated with adolescent obesity and overweight in Korea were included in the introduction. (2,3) I’ve used references that you suggested and clarified the discussion section

The planning and execution of health-policies associated with the prevention and management of obesity in adolescence are mainly developed by the central government, represented by the Ministry of Health and Welfare. According to the first Master Plan for Student Health Promotion (2019~2023), the local education offices should prepare community-appropriate strategies by considering the conditions and characteristics of each region, while schools should manage local communities and related institutions [39]. It seems adolescent obesity issue is entrusted only to the local system and the central government put little effort to consider about the relevance and liability of applying the policies to each region [40]. In order to prevent and manage adolescent obesity in the aspect of health inequality between regions, the followings should be implemented. First, a development of valid adolescent health monitoring system should be prepared. Unlike the KYRBS which is conducted and managed by the Centers for Disease Control Agency, the Ministry of Education deals with the NSHE. Taking the strength and weakness of each data sources into consideration, both organizations should deeply reconsider about the data collection process and revise if necessary. By doing so, sufficient and valid evidence would be generated at the national and regional levels. Second, establishment of a governance to solve adolescent obesity problem is necessary. Centered by the Ministry of Health and Welfare and the Ministry of Education, health-related sectors, including the Ministry of Environment, local education offices, and school districts, should cooperate to resolve the problem [39]. The central government could allocate more grants to the most obese regions so that establishment of relevant related policies to help produce positive results, including the enhancement of health-related behaviors and the reduction in diseases, disabilities, or unexpected deaths in adulthood can take a place. Some effective policy options in reducing adolescent obesity suggested by an Australian study included nutrition education, physical education, and parental involvement in such activities [41]. It would be also possible for the government to develop daily physical activity guidelines based on the obesity level of each region and to incentivize schools which meet the proposed guidelines and show improvements [42].

Some references are old (Ref# 4,5,8)- Please update.

  • The above references were updated. Thank you for noticing.

Reviewer 3 Report

This study aimed to compare two nationally representative data sources of Korean adolescent obesity and overweight prevalence at the regional levels.The authors demonstrated that the two different nationally representative data could be used in the examination of the adolescent obesity burden at the regional level.

The main problem with the paper is that there is a substantial difference in the methods used for measuring weight and height. In the KYRBS, students were asked to self-report their latest height and weight while, in the NSHE, they were directly measured. It is clear that this difference could be the cause of substantial discrepancies between the two data.

The importance of analysing the geographical distribution of adolescent obesity and overweight status is not very clear. The introduction and discussion of the paper are confusing. The way the data are presented and described it appears that this paper should be published in a journal dealing with epidemiology. 

Author Response

  • Thank you for your valuable comments and I am sorry that this study did not fully satisfy you. We, the authors of this study, are totally aware of the fact that the measurement methods used to collect height and weight data of adolescents were not the same between the two data sources. Therefore, we designed a study to compare regional distribution of obesity status by comparing relative rankings between the obesity and overweight prevalence estimated using the KYRBS (self-reported) and the NSHE (directly measured). We focused on the following issues when dealing with this study; 1) the value of examining obesity and overweight status of adolescents at the regional levels considering health inequality issue, which is getting serious in Korea and 2) reconsideration of the quality of the available data in order to prove what to make out of it. We’ve made some improvements based on the reviewers’ comments, especially in the introduction and discussion section. I would appreciate if you could re-review the revised version if you don’t mind. Thank you.

Reviewer 4 Report

Dear Authors,

Thank you for your manuscript.

The topic is important and relevant to the current situation of increasing adolescents overweight and obesity. Valid national health and lifestyle monitoring systems are extremely important.
My section-specific comments are provided below.

Introduction. Trends in prevalence in adolescent overweight and obesity in South Korea should be described in the Introduction. Moreover, in my opinion, at the end of the Introduction, the readers should be introduced that the comparison of the prevalence of overweight and obesity will be based on self-reported vs objectively measured sources (as well as in the abstract).

Methods. For the international reader following points not provided in the description of the national health examinations would be interesting:
KYRBS: Were environmental, family-related and social support factors involved in the survey?
NSHE: What was the age of the students? Age in grades might be different across countries. What information was involved in the survey questionnaire form?
In the description of statistical analysis, it is mentioned that some data were excluded from the analysis due to inaccuracies in age and weight. How many?

Results. It should be indicated that data in Figures 1 and 2 are based on correlations. Also, correlation coefficients (r) provided together with the p-values on the graphs would provide more clarity.
Together with the comments on Tables 2 and 3, it is recommended to indicate the prevalence of overweight and obesity based on each source (KYRBS, NSHE) and year (2018, 2019): e.g. from ... to according to the city. Also, possible reasons of obesity prevalence differences across cities should be discussed as they are really impressive, e.g., KYRBS, boys, 2019 from 21.8% in JJ down to 10.2% in SJ.  

Author Response

Dear Authors,

Thank you for your manuscript.

The topic is important and relevant to the current situation of increasing adolescents overweight and obesity. Valid national health and lifestyle monitoring systems are extremely important.
My section-specific comments are provided below.

  • Thank you for your time and valuable comments.

Introduction. Trends in prevalence in adolescent overweight and obesity in South Korea should be described in the Introduction. Moreover, in my opinion, at the end of the Introduction, the readers should be introduced that the comparison of the prevalence of overweight and obesity will be based on self-reported vs objectively measured sources (as well as in the abstract).

  • As you recommended, recent obesity and overweight prevalence change in Korea was added in the introduction section.

Similarly, recent obesity and overweight trends in Korea are also increasing from 15.3% in 2007 to 23.7% in 2017 [2].

  • According to your suggestion, the following statement was inserted in the last paragraph of the introduction section.

In the KYRBS, students were asked to self-report their latest height and weight while, in the NSHE, they were directly measured.

Methods. For the international reader following points not provided in the description of the national health examinations would be interesting:
KYRBS: Were environmental, family-related and social support factors involved in the survey?
NSHE: What was the age of the students? Age in grades might be different across countries. What information was involved in the survey questionnaire form?

In the description of statistical analysis, it is mentioned that some data were excluded from the analysis due to inaccuracies in age and weight. How many?

  • In the KYRBS, questions associated with health equity are included. The questions are involved with family members, residential status, parents’ educational levels, parents’ nationality, subjective academic performance, subjective socioeconomic status, relationship with family/friends/teachers, average allowance per week, etc. To help the readers, the fact that the survey questionnaires include such factors was additionally mentioned. Thank you.

Many researchers examining obesity-related problems in adolescence prefer using the KYRBS [26-28], since it provides structurally designed questionnaires regarding health behaviors and socioenvironmental factors.

  • The age ranged from 6 to 18 for the NSHE. As mentioned in the manuscript, data from elementary school students were excluded from the analysis to compare obesity and overweight status of adolescents with that of the KYRBS data, which is conducted for students in junior high and high schools. To help readers, the following correction was made.

To compare obesity and the overweight status data of adolescents with that of the KYRBS data, data from elementary school students (1st – 6th grades aged from 6 to 11) were not analyzed in this study.

  • I am sorry about the inconvenience. As described in the methods, the number of students who participated in the 2018 wave was 60040 and 57303 students participated in the 2019 wave for the KYRBS. For the NSHE, the number of students who checked physical developmental status in the 2018 wave was 107954 and 104380 students in the 2019 wave. Data from elementary school students were excluded from the NSHE (39091 for the 2018 wave and 39084 for the 2019 wave) as described in the manuscript. In addition, the number of students excluded from the KYRBS analysis due to inaccuracies in age and weight/height is 3260 (1704 for the 2018 wave; 1556 for the 2019 wave) and that of the NSHE is 150 (72 for the 2018 wave; 78 for the 2019 wave). Therefore, as described in the results, the total number of the KYRBS and NSHE participants included in the analysis were 114083 (58336 for the 2018 wave; 55747 for the 2019 wave) and 134079 (68861 for the 2018 wave; 65218 for the 2019 wave), respectively. To help readers, a figure (figure 2) showing the final participants was additionally provided. Please check. (Originally indicated figures 2 and 3 were revised to figures 3 and 4 accordingly.)

Results. It should be indicated that data in Figures 1 and 2 are based on correlations. Also, correlation coefficients (r) provided together with the p-values on the graphs would provide more clarity.

  • According to your suggestion, figures and figure legends were revised. Please check.

Together with the comments on Tables 2 and 3, it is recommended to indicate the prevalence of overweight and obesity based on each source (KYRBS, NSHE) and year (2018, 2019): e.g. from ... to according to the city. Also, possible reasons of obesity prevalence differences across cities should be discussed as they are really impressive, e.g., KYRBS, boys, 2019 from 21.8% in JJ down to 10.2% in SJ.  

  • Thank you for your suggestion. According to your suggestion, some differences in adolescent obesity prevalence across regions were emphasized in the discussion section. Since there are some overlapping in 95% CIs, we’ve only focused on the least and most obese regions and pointed out that the 95% CIs of these figures did not overlap. We also believe that this result is impressive since this might indicate worsening of health inequality among adolescents. Thank you for noticing.

Possible reasons of obesity prevalence differences across regions may include differences in their health behaviors. Taking JJ for example, it is previously reported that children and adolescents in Jeju showed lower physical inactivity than students in Seoul due to shortage of public transportation so that many of them go to school with help from their parents, giving a ride [25-28]. The following paragraph was added in the revised version. Please check. Thank you.

The levels of adolescent obesity prevalence estimated from the KYRBS and NSHE both showed that there were some differences across regions. Regardless of data sources used, for both boys and girls, there were no overlapping between the 95% CIs of the least and most obese regions. Obesity prevalence of boys estimated from the 2019 KYRBS ranged from 10.0 (7.2-13.2, SJ) to 21.8 (18.0-25.6, JJ), and that identified from the NSHE ranged from 15.3 (13.3-17.3, GJ) to 23.0 (19.9-26.1, JJ). Similarly, obesity prevalence of girls estimated from the 2019 KRYBS ranged from 4.7 (2.6-6.9, SJ) to 10.6 (7.4-13.8, GW), and that identified from the NSHE ranged from 12.3 (10.1-14.5, SJ) to 19.9 (16.7-23.1, JJ). This suggests possible differences in adolescents’ health behaviors by region.

Round 2

Reviewer 2 Report

Dear Authors,

The paper has significantly improved by these revisions. However, the novelty of the study could be improved. The author's response to my comment is logic but should be clearly mentioned at the end of introduction or included as a limitation of the study. In the statistical analysis, authors should mention if the data meet the assumptions of a normal distribution because the Spearman correlation coefficient is a non-parametric test which measures the strength of association between two variables. In addition, the conclusion section should contain more information to qualify as a stand alone section.

Reviewer 3 Report

The authors have made some changes to the paper. The main problem remains. The study presents the comparison of two different nationally data of Adolescent Obesity and Overweight Status in Korea from 2018-2019. I don't think this is interesting data for a health journal but rather for one dealing with epidemiology. 

Reviewer 4 Report

Dear Authors,

Thank you. I appreciate your taking into account all my comments.

This manuscript is a resubmission of an earlier submission. The following is a list of the peer review reports and author responses from that submission.

Round 1

Reviewer 1 Report

Please adjust the x and y axis for figure 3 (crop) so that the correlation is better visible.

Other than this I do not have other comments as the study is made on a large enough cohort and, as obesity is a problem with which we are being confronted today more and more, I consider that this manuscript has a high importance at national level.

Reviewer 2 Report

In this paper, the authors used two datasets to estimate the prevalence of adolescents obesity and overweight in different regions of South Korea. These numbers should be an important reference for policymaking, which is also the main purpose of this paper. 

As the authors pointed out that using two datasets is a novelty of this paper while most other papers only used one data source. However, it is also the issue of this paper. From both Table 2 and 3, especially Table 2, we can see the big difference of prevalence estimation between the two datasets. For most pairs, there is no overlap of 95% CI of two estimations. They are significantly different. Accurate estimation is necessary for policymaking. Although the authors discussed limitations of the two datasets in 4.1, we do not know which one we should use. For example, the obesity rate of girls in SU is 6.9% in KYRBS in 2018, which is not a big issue as the authors used 95th percentile as cutoff. The estimation is 12.9% in NSHE, which is a number we should pay attention during the policy making. The authors should discuss how other researchers or policymakers could use the results in this paper. 

Correlation analysis only shows the correlation between two analysis.It does not mean the two variables are similar to each other. Two prevalence like 5%, 6%, 7%, 8% and 25%, 26%, 27%, 28% have high correlation but are not similar. In this study, the best scenario is the estimations from two datasets are similar. We can see that the correlation between two datasets is higher for estimation of obesity prevalence than the overweight prevalence. However, the estimation of prevalence is more similar between two datasets in overweight than in obesity. 

In Figure 1, the regions have been divided into 4 categories. Is prevalence similar within each category? For example, the obesity rate is high in all metropolitan cities.

The authors used weighted percentage. I am not sure how the weighted percentage was calculated, like weighted average. In that case, what kind of weight was used in the paper?

Line 191 and some other lines, 4.0%p and 4.7%p. What does 'p' indicate?

Figure 3, what does "*" and "**" indicate in this figure?

There are many typos in the paper. The authors should proofread it carefully before the next submission.  A few typos:

Line 54: "Meanwhile, news articles usually provide". new articles?

Line 77: "the levels of fiscal availabilities are also difference". different

Line 82: "by a variety of organizations". by variety of organizations

Line 95: "sexual education". sex education.

Reviewer 3 Report

The manuscript contains new and significant information adequate to justify publication. The prevention of obesity is an issue of great importance and very topical. The importance of analyzing the geographical distribution of adolescent obesity and overweight status in Korea is emphasized by this study. Autor’s expected that the study results could be used as scientific evidence in the establishment of adolescent obesity prevention and management policies. I agree with the authors that this could be a useful method. The authors clearly presented the limitations of this comparative study.

Reviewer 4 Report

The manuscript deals with the use of data from Korean national programs tracking various information about children and adolescents.

In the present work, the authors compare 2 huge files, which are certainly informationally valuable, but I explicitly mind that if the authors want useful and clinical output, why do they compare within the sets of rankings and rank difference from two different files, which are (for methodological reasons) comparable only relatively. Even if the whole population gains / loses weight, it can still be first in order and last in order. I don't see any point in that. Especially if within the KYRBS file w% decreases year-on-year and the NSHE file w% year-on-year. From a epidemiological / clinical point of view, it is much more useful to observe dynamic changes within regions, the relationship between overweight and obesity, because the results show that in some regions w% with overweight decreased but with obesity increased, but in others the increase in obesity copies the increase in overweight.

I do not see the usefulness of this article.